# PROMPT-GUIDED DYNAMIC NETWORK FOR IMAGE SUPER RESOLUTION

## ABSTRACT

Existing single image super-resolution (SISR) methods learn the convolutional kernel solely from a single image modality. However, the SR performance is limited by the diversity of input modality and the insufficient image-level information in low-resolution images. In this paper, we seek to use multi-modal prompts (texts or images) to assist existing SR networks to learn more discriminative features, leading to superior SR performance. To this end, we develop the Dynamic Correlation Module in a plug-and-play form for existing SR networks, which learns meaningful semantic and textural information from multi-modal prompt embeddings extracted from a large-scale vision-language model (such as CLIP). Specifically, Spatially Multi-Modal Attention Module is proposed to generate the pixel-wise cross-modal attention mask which would highlight the interest regions given certain prompts. Moreover, to the best of our knowledge, we are the first ones that introduce multi-modal prompts into convolutional kernel estimation which can better handle spatial variants and retain cross-modal relevance. Extensive experiments and ablation studies demonstrate the effectiveness of the proposed Dynamic Correlation Module which exploits the discriminative prompt features to recover realistic high-resolution images, elevating existing SR performance by a notable gap.

## 1 INTRODUCTION

Single image super-resolution (SISR) aims to recover high-resolution (HR) images given their low-resolution (LR) counterparts. As a long-standing low-level computer vision problem, SISR has been investigated for decades (Sun et al., 2008; Zhang et al., 2012; Yang & Yang, 2013) and can be applied to many downstream tasks, such as surveillance and medical imaging (Trinh et al., 2014; Pang et al., 2019).

Existing SISR methods can be categorized into optimization-based methods (Mairal et al., 2009; Dong et al., 2012; Elad & Aharon, 2006; Aly & Dubois, 2005; Riegler et al., 2015; He & Siu, 2011; Tipping & Bishop, 2002) and learning-based methods (Saharia et al., 2022; Wang et al., 2018b; Lim et al., 2017; Zhang et al., 2021; Ma et al., 2022; Rombach et al., 2022; Bell-Kligler et al., 2019; Gu et al., 2019; Ledig et al., 2017; Zhang et al., 2018b; Shi et al., 2016; Sun et al., 2023; Wang et al., 2023; Gao et al., 2023; Gou et al., 2023). For optimization-based methods, probabilistic models are elaborately designed to simulate the HR degradation process. However, tedious optimization procedures are required to super-resolve low-resolution images, which does not meet the real-time need. Since the pioneering work SRCNN (Dong et al., 2014), learning-based SR methods (Saharia et al., 2022; Wang et al., 2018b; Lim et al., 2017; Zhang et al., 2021; Ma et al., 2022; Rombach et al., 2022; Bell-Kligler et al., 2019; Gu et al., 2019; Ledig et al., 2017; Zhang et al., 2018b; Shi et al., 2016) have brought prosperous progress and surpass optimization-based SR methods by a huge gap. Learning-based methods learn the specific mapping function from LR-HR image pairs with Convolutional Neural Networks (CNNs), which implicitly learn the convolutional kernels solely from the image training datasets in an end-to-end manner. However, we argue that a single LR image may not provide enough information to recover the HR image details, especially when the scale factor is large. Although some non-blind SR methods bring blur kernel information into the super-resolve procedure to remedy this problem, the SR performance is still limited due to the static inference phase (Xu et al., 2020). Nowadays, it is still an open problem that the convolutional kernels learned by existing SR methods are not robust enough against various degradations (Xu et al., 2020;

Kim et al., 2021), which may cause the super-resolved images to be the statistical average of possible HR solutions (Ma et al., 2020).

To address the aforementioned issues, this paper proposes a new *Prompt-guided Dynamic Network (PDN)* to introduce powerful multi-modal representations to existing SR frameworks, where the prompts reflect semantic descriptions of the input LR image. In practice, such descriptions may be texts (e.g., captions or textual descriptions) or related images (e.g., an image similar to the LR input or another augmented view). We seek to obtain convolutional kernels by such multi-modal semantic descriptions to keep the sensitivity to spatially-variant degradations, which may boost the SR performance. To this end, we propose a *Dynamic Correlation Module (DCM)*, which is the first attempt that utilizes multi-modal prompts to adjust convolutional kernels. Contrary to the existing static SR networks, PDN can adaptively alter the weights of convolutional kernels for specific spatial regions depending on the semantic description in the inference phase, resulting in superior SR performance.

Specifically, the proposed DCM is made up of Spatially Multi-Modal Attention Module and Prompt-Guided Dynamic Convolution Module. Given the textual or visual semantic descriptions, we first derive the prompts with a large-scale pre-trained model (such as CLIP (Radford et al., 2021)), which is a natural solution so that our model not only obtains more discriminative features for downstream tasks but also supports mixed input of images and text as prompts (Gu et al., 2021; Kuo et al., 2022). Then, the spatially multi-modal attention predicts the interest regions in space that are semantically related to the prompts by measuring the similarity between image features and prompt embeddings. Based on that, the Prompt-Guided Dynamic Convolution Module derives appropriate convolutional kernels from the kernel bank for feature transformation according to the prompts. These two techniques promote each other and jointly enable SR networks to learn meaningful semantic information and enrich the details, leading to an elevation of SR performance. In addition, the proposed Spatially Multi-Modal Attention Module and Prompt-Guided Dynamic Convolution Module are flexible and versatile and can be conveniently incorporated into various SR frameworks as a plug-and-play module.

We conduct extensive experiments on four popular benchmark datasets, Set5 (Bevilacqua et al., 2012), Set14 (Zeyde et al., 2012), Urban100 (Huang et al., 2015), and Celeba-HQ (Karras et al., 2018). Experimental results demonstrate the superiority of the proposed technique in improving existing SR methods. We incorporate our modules into many state-of-the-art SR networks and elevate the quantitative results by a notable gap. For example, with text descriptions, PDN improves PSNR up to 0.11 and improves SSIM up to 0.013 over state-of-the-art SR methods at the scale factor of $\times 4$.

The main contributions are as follows:

- This paper proposes a novel Prompt-guided Dynamic Network (PDN) which introduces powerful multi-modal representations to existing SR frameworks. PDN is capable of learning meaningful semantic information from prompts, whose key component is the Dynamic Correlation Module (DCM). The technical contributions of DCM include **1)** a Spatially Multi-Modal Attention Module and **2)** a Prompt-Guided Dynamic Convolution Module. Such components enable PDN to learn meaningful semantic and texture information from multi-modal prompts, making full use of both LR images and prompts.

- To the best of our knowledge, we are the first to introduce multi-modal prompts into convolutional kernel estimation for feature transformation, leading to a better capability of modeling cross-modal coherence and spatial variations.

- We conduct comprehensive analyses of the effectiveness of the proposed modules, demonstrating the significance of introducing multi-modal information into SR models, which may inspire further research.

## 2 RELATED WORKS

Before the deep-learning era, optimization-based methods (Mairal et al., 2009; Dong et al., 2012; Elad & Aharon, 2006) dominate the SR field. Probabilistic models are established to simulate the HR degradation process. However, tedious optimization procedures are required to super-resolve low-resolution images, which does not meet the real-time need. SRCNN (Dong et al., 2014) first

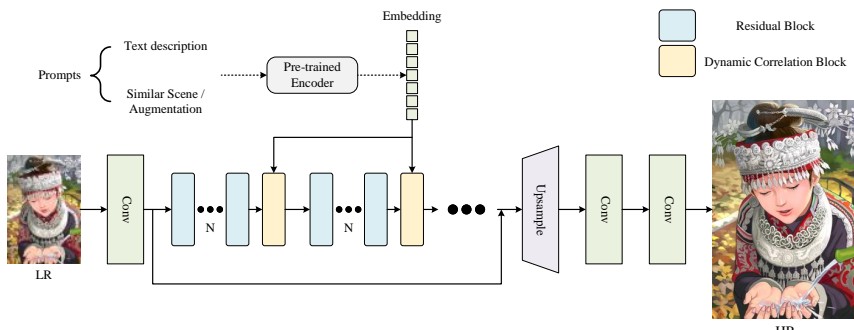

Figure 1: Diagram of the proposed Prompt-guided Dynamic image super-resolution Network (PDN).

introduces a three-layer CNN to SR and shows remarkable improvements over optimization-based methods. In recent years, with the development of residual learning (He et al., 2016), SRRes-Net (Ledig et al., 2017), EDSR (Lim et al., 2017), RCAN (Zhang et al., 2018b), RDN (Zhang et al., 2018c), and ESRGAN (Wang et al., 2018b) take SR performance to the next level. These methods usually stack several residual blocks and expand the SR network, posing impressive SR results when handling bicubic degradation. Additionally, to enhance the robustness of SR models for various degradations (such as isotropic/anisotropic Gaussian blur), a series of representative SR methods (Bell-Kligler et al., 2019; Gu et al., 2019; Huang et al., 2020) split the problem of SR into kernel-estimation and non-blind SR. By estimating accurate blur kernels and unfolding the kernel information into SR networks, non-blind SR methods retain robustness to variant degradations. Nevertheless, most non-blind SR methods are trained on synthetic datasets, which implicitly rely on "fixed" and "ideal" blur kernels, leading to high sensitivity to the accuracy of kernel estimation. Thus, small deviations from the ground-truth kernel may lead to significant performance drops.

## 2.1 GUIDED IMAGE SUPER RESOLUTION

The ill-posed nature of SISR makes it difficult to get better SR results just solely depending on LR images. Recently, SPSR (Ma et al., 2020), FSRNet (Chen et al., 2018), DeepSEE (Buhler et al., 2020), and SFTGAN (Wang et al., 2018a) claim that the internal/external priors (e.g., image edges, semantic segmentation maps, and facial landmark heatmaps) can significantly boost the SR performance. For example, SPSR adds an additional gradient estimation network alone with an existing SR network. Compared with the baseline SR network ESRNet (Wang et al., 2018b), SPSR alleviates the issue of geometric distortion which commonly appears in GAN-based SR methods. Generally, SPSR utilizes gradient information to guide the super-resolve process. It does not require prompts of different modalities. Furthermore, inspired by the success of text-to-image synthesis, TGSR (Ma et al., 2022) regards image SR as text-guided detail generation, which is the first attempt that makes use of multi-modal prompts in SR. The proposed text attention module (TAM) incorporates text embeddings and features and highlights the regions of interest corresponding to each input word. Nevertheless, these guided SR methods either limit the diversity of guidances or fail to investigate the full potential of guidances, leading to inferior quantitative results compared with existing SR methods.

## 3 PROPOSED METHODS

### 3.1 MOTIVATION

Conventional single image super-resolution (SISR) takes low-resolution images $I^{LR}$ as input and generates super-resolved images $I^{SR}$ given its high-resolution counterparts $I^{HR}$. For blind SR, let $G_\theta$ denote the SR network with parameters $\theta$, we have $I^{SR} = G_\theta(I^{LR})$. And for non-blind SR with additional input blur kernel $k$, we have $I^{SR} = G_\theta(I^{LR}, k)$. The blur kernel is usually compressed by the dimensionality stretching strategy (Zhang et al., 2018a). Typically, an estimator $E$ is utilized to predict the blur kernel information given $I^{LR}$ as input (Bell-Kligler et al., 2019; Gu et al., 2019),

namely $k = E(I^{LR})$. Obviously, both LR images and blur kernels are solely derived from image modality for conventional (both blind and non-blind) image SR.

Unlike conventional image SR, the prompt-guided image SR pursues to explore the potential of multi-modal prompts for SR. We use $p$ to denote the SR prompt, which can either be the caption of the HR image or an image with scenes that are similar to the HR image. Intuitively, we desire the SR prompt to serve as the global description by providing a variety of decisive information about the interested objects (such as the gender of the person or the color of the sky), so that the SR network can better locate the most important objects and pay more attention to these areas, which will boost the final SR performance. Furthermore, conventional image SR methods usually suffer significant performance drops when dealing with different degradations (Gu et al., 2019; Zhang et al., 2020). This is mainly due to the static network architecture (Xu et al., 2020), which lacks the capability to handle cross-image or spatial variations. To solve this problem, we propose a novel mechanism to generate the dynamic convolutional kernels in order to make the SR model capable of utilizing *multi-modal prompts*, which clearly distinguishes our method from the prior works that generate the kernels solely according to the extracted image features (Xu et al., 2020; Chen et al., 2020).

Specifically, we propose *Prompt-guided Dynamic Network (PDN)*, which is composed of regular SR network components, and our proposed *Dynamic Correlation Module (DCM)*. In the following subsections, we first give a brief description of PDN, then elaborate on DCM, where the critical designs include *Spatially Multi-Modal Attention Module* and *Prompt-Guided Dynamic Convolution Module*.

## 3.2 PROMPT-GUIDED DYNAMIC NETWORK

We present the Prompt-guided Dynamic Network (PDN) to introduce powerful multi-modal representations into the super-resolution procedure. The sketch of PDN is shown in Figure 1, which consists of the multi-modal pre-trained encoder, Dynamic Correlation Modules, and several residual blocks. For the multi-modal prompt encoder, we utilize CLIP for its generalization ability and robustness. We adopt the residual in residual (RIR) block (Zhang et al., 2018b) as the basic residual block. The optimization objective of PDN is to minimize the $L_1$ loss function

$$L = \mathbb{E}_{I^{LR}} ||G_\theta(I^{LR}) - I^{HR}||_1 . \tag{1}$$

## 3.3 DYNAMIC CORRELATION MODULE

A Dynamic Correlation Module consists of Spatially Multi-Modal Attention Module and Prompt-Guided Dynamic Convolution Module. It is lightweight and can be easily plugged into existing SR networks.

**Spatially Multi-Modal Attention Module**. Given the prompt encoder $Q$ and prompt $p$, we first obtain the prompt embedding $f_p = Q(p) \in \mathbb{R}^{1 \times d}$, where $d$ is the pre-defined dimension of prompt embedding. We use $f_{LR} \in \mathbb{R}^{c \times h \times w}$ to denote the intermediate feature of the LR image that is produced by a specific layer of the model, where $c$ is the number of channels. To generate the cross-modal attention mask, $f_{LR}$ is projected into $d$ channels by a $1 \times 1$ convolution. Then we reshape the projected $f_{LR}$, which is $C_{1 \times 1}(f_{LR}) \in \mathbb{R}^{d \times h \times w}$, into a matrix $f'_{LR}$ for the following attention computations, so $f'_{LR}$ is a $(hw) \times d$ matrix.

Then we apply $L2$ normalization to both $f'_{LR}$ and $f_p$. The attention mask $M_c \in \mathbb{R}^{(hw) \times 1}$ is given by matrix multiplication of the normalized $f'_{LR}$ and $f_p$. That is

$$M_c = S((\frac{f'_{LR}}{||f'_{LR}||_2} \cdot (\frac{f_p}{||f_p||_2})^T) \cdot \zeta), \tag{2}$$

where $\zeta$ is a learnable scaling factor initialized as 50, and $S$ denotes the Sigmoid function.

Intuitively, a specific value in the multi-modal attention mask describes the degree of similarity between the intermediate feature at the corresponding spatial location and the prompt embedding so that the regions of interest will be highlighted, compared to those not involved in the prompt. Besides, considering that the values of both prompt embedding and image feature are suppressed by $L_2$ normalization, we scale the mask by a learnable factor $\zeta$ to keep a relatively large variance.

Such a multi-modal attention mask enables the model to attend to the features adaptively, depending on the prompts, which is vital for comprehending the semantics encoded in the features. Then the multi-modal attention mask $M_c$ is element-wisely multiplied to $f'_{LR}$. The result, which is referred to as the enhanced feature $f_{enhanced} \in \mathbb{R}^{(hw) \times d}$, will be fed into the Prompt-Guided Dynamic Convolution Module. That is

$$f_{enhanced} = M_c \odot f'_{LR}, \tag{3}$$

where $\odot$ is Hadamard (element-wise) multiplication. Of note is that $M_c$ should be broadcast (i.e., replicated) into the same shape as $f'_{LR}$ to be more formal, but we omit such an obvious notation for brevity.

**Prompt-Guided Dynamic Convolution Module** takes the enhanced feature and prompt embedding as inputs. Intuitively, this module leverages prompt embedding to guide the convolution operation for a better prompt-relevant feature representation. Drawn insight from the success of dynamic convolution, the forward pass of the prompt-guided dynamic convolution module is demonstrated as follows.

$$f_o = (\sum_{k=1}^{n} \pi_k(f_p) W_k) f_{enhanced} + \sum_{k=1}^{n} \pi_k(f_p) b_k,$$

$$s.t., 0 \leq \pi_k(\cdot) \leq 1, \sum_{k=1}^{n} \pi_k(\cdot) = 1 \tag{4}$$

First, the prompt embedding $p$ is projected into an $n$-dimensional attention vector $\pi$ through a two-layer MLP, where $n$ is the pre-defined number of trainable basis kernels in the kernel bank (which is set to $n = 4$ by default). Then, we derive the kernel as a sum of the $n$ basis kernels weighted by the softmax of $\pi$ together with the bias, as shown in Figure 2. The result of the convolution on the enhanced feature with the weighted kernel is the output of the DCM.

**Remark** Compared to the conventional dynamic convolution (e.g., CondConv (Yang et al., 2019) for image recognition), the main difference is that we use the prompt embedding instead of the image feature to generate the dynamic attention weights $\pi$. The reason for doing so is two-fold. First, the conventional mechanism derives the attention weights based on the image feature only, which is not suitable for modeling cross-modal coherence. In contrast, we generate dynamic weights by the prompt embedding, which naturally ensures the image-prompt correlation. Second, conventional dynamic weights are usually "averaged" in the SR task (Chen et al., 2021), which means the basis and the weighted kernels tend to end up with little difference. This is mainly caused by the relatively low

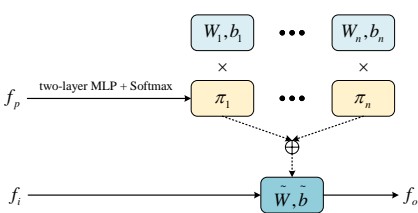

Figure 2: Comparison between conventional dynamic convolution (a), and the prompt-guided dynamic convolution (b). $\pi$ denotes the dynamic weights.

variance of the feature's global statistics, which limits the potential of conventional dynamic convolution. On the contrary, the prompts embeddings extracted by a large-scale vision-language model have high variance and sparse distributions (Khandelwal et al., 2022; Tevet et al., 2022), which can guide the weighted kernels to learn more discriminative patterns.

## 3.4 INTEGRATION INTO EXISTING SR METHODS

For a fair comparison, we integrate the proposed Dynamic Correlation Module into state-of-the-art PSNR-oriented image SR methods: EDSR (Lim et al., 2017), RDN (Zhang et al., 2018c), and RCAN (Zhang et al., 2018b). These three methods utilize stacks of elaborately designed residual blocks to generate HR images with plausible details. For EDSR, we add one Dynamic Correlation Module after each residual block. For RCAN, we add three Dynamic Correlation Modules in each residual in the residual (RIR) block (with identical intervals). For RDN, we add one Dynamic Correlation Module after each residual dense block (RDB). For simplicity, we refer to these three upgraded networks as EDSR+, RDN+, and RCAN+ in the following sections.

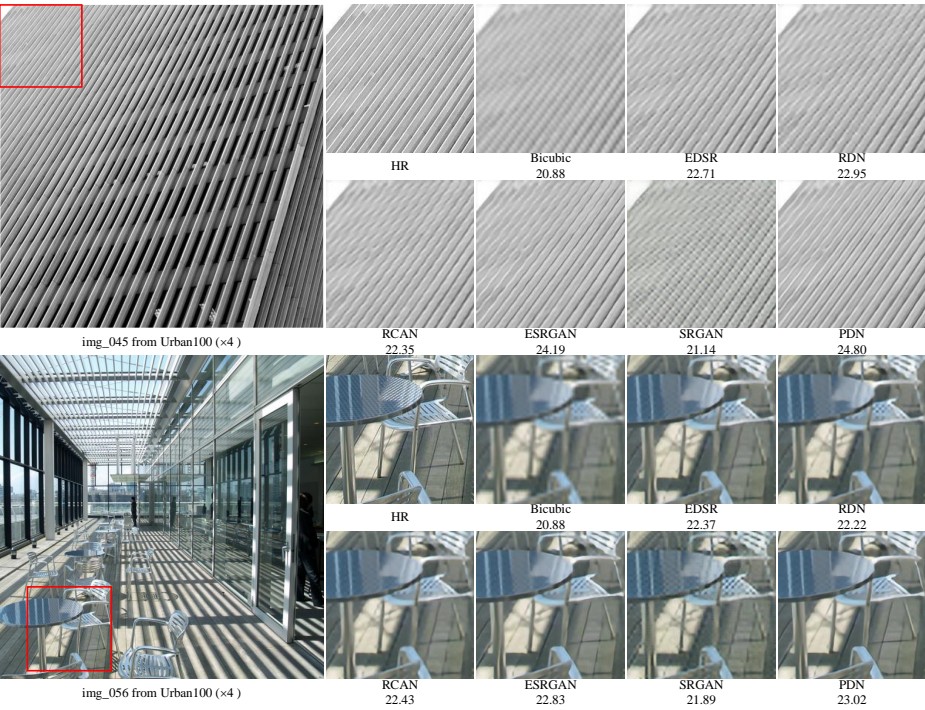

Figure 3: Visual comparison with state-of-the-art SR methods on Urban100 dataset with a scale factor of ×4. Best viewed on screen.

# 4 EXPERIMENTS AND ANALYSIS

## 4.1 DATASETS AND TRAINING DETAILS

For a comprehensive and impartial comparison, we use COCO (Lin et al., 2014) and FFHQ (Karras et al., 2019) datasets for training and the commonly used benchmarks for evaluation: Set5 (Bevilacqua et al., 2012), Set14 (Zeyde et al., 2012), Urban100 (Huang et al., 2015), and Celeba-HQ (Karras et al., 2018).

We train for 100 epochs with a batch size of 16, an initial learning rate of $10^{-4}$, cosine annealing schedule (Loshchilov & Hutter, 2016), and an ADAM optimizer (Kingma & Ba, 2014) with $\beta_1 = 0.9$, $\beta_2 = 0.999$, and $\epsilon = 10^{-8}$. The temperature coefficient of the softmax function in the Prompt-Guided Dynamic Convolution Module is set to 34 and is decayed by 3 every 10 epochs. We use PSNR and Structure Similarity (SSIM) as the evaluation metrics.

In the training phase, we first filter out images in the COCO dataset with the longest edge of less than 512. Then we crop the raw images into sub-images in the shape of $320 \times 320$. After that, we blur the HR images by isotropic Gaussian kernel with $\sigma = 1.3$ and downsample by bicubic interpolation to obtain the corresponding LR images. As the inputs to the model are $48 \times 48$ patches cropped from the LR images, the corresponding HR patches should be $48s \times 48s$, where $s$ denotes the scale factor. As for the captions of the COCO dataset, we randomly choose one caption for each image and use the CLIP text encoder to get its prompt embedding, which is finished in advance as a preprocessing procedure so as to accelerate the training. Considering that FFHQ, Set5, Set14, and Urban100 datasets have no annotations, we use an augmented view as the semantic description. Specifically, we send the horizontally flipped image to the CLIP image encoder to get the prompt embedding.

## 4.2 COMPARISON WITH STATE-OF-THE-ARTS

We compare the proposed PDN with state-of-the-art image SR methods: RCAN (Zhang et al., 2018b), RDN (Zhang et al., 2018c), EDSR (Lim et al., 2017), SRGAN (Ledig et al., 2017), ES-

Table 1: Quantitative comparison with state-of-the-art SR methods. The best performance is highlighted in **bold**.

| Method | Scale | COCO | | | Scale | FFHQ |
| | | Set5 PSNR/SSIM | Set14 PSNR/SSIM | Urban100 PSNR/SSIM | | Celeba-HQ PSNR/SSIM |
|---|---|---|---|---|---|---|
| RCAN | x2 | 37.33/0.938 | 33.26/0.899 | 31.19/0.907 | x8 | 32.94/0.877 |
| RDN | | 37.06/0.936 | 33.02/0.896 | 30.72/0.898 | | 32.99/0.878 |
| SRGAN | | 27.83/0.843 | 25.83/0.768 | 23.70/0.752 | | 28.28/0.839 |
| ESRGAN | | 37.15/0.937 | 33.14/0.898 | 31.10/0.901 | | 32.46/0.868 |
| EDSR | | 37.25/0.935 | 33.16/0.898 | 31.01/0.902 | | 32.43/0.870 |
| PDN | | **37.46/0.939** | **33.37/0.900** | **31.45/0.907** | | **33.27/0.881** |
| RCAN | x4 | 30.26/0.825 | 27.46/0.744 | 24.34/0.707 | x16 | 28.74/0.826 |
| RDN | | 30.55/0.836 | 27.60/0.751 | 24.55/0.718 | | 28.57/0.820 |
| SRGAN | | 27.45/0.810 | 24.81/0.706 | 22.54/0.688 | | 27.88/0.812 |
| ESRGAN | | 31.39/0.856 | 28.12/0.768 | 25.55/0.754 | | 28.85/0.828 |
| EDSR | | 30.74/0.842 | 27.75/0.756 | 24.74/0.726 | | 28.55/0.824 |
| PDN | | **31.47/0.857** | **28.20/0.769** | **25.66/0.759** | | **29.18/0.833** |

RGAN (Wang et al., 2018b), and SPSR (Ma et al., 2020). Specifically, we make a comparison on the COCO dataset with scale factors of ×2 and ×4, and a comparison on the FFHQ dataset with scale factors of ×8 and ×16. Results of PSNR and SSIM values are presented in Table 1. Note that RDN does not provide ×8 and ×16 models in the official source code, so we add additional up-sampling blocks for ×8 and ×16 experiments. At each row, the best results are highlighted in bold. Impressively, PDN achieves the best PSNR and SSIM performance on all of the datasets and the gap increases with large-scale factors or large test datasets. In summary, the quantitative results demonstrate the superiority of PDN in perceptual quality.

We also conduct a visual comparison on the Urban100 dataset with a scale factor of ×4. From Figure 1, we see that our results are more natural and realistic than other methods. Taking the image "img_056" for example, it can be observed that most of the existing methods cannot recover the wooden floors with reasonable edges and details. In contrast, PDN can better alleviate the blurring artifacts and restore more details. And as for image "img_045", all the other methods are prone to generate blurry edges, which contradict the HR image, while the structure of our result is clear with little distortions. The visual comparison shows that the proposed Dynamic Correlation Module can enhance existing SR methods to learn more structural and textural information from prompts, which helps generate perceptual-realistic SR images. More visual results can be found in the supplementary material.

## 4.3 COMPARISON WITH TGSR

TGSR Ma et al. (2022) regards SISR as a text-guided semantic image detail enhancement problem, which aims to generate high-resolution images that match the text descriptions in a coarse-to-fine process. As the authors did not release the official code, we reproduce the experiments with the same setup as reported in the paper (Ma et al., 2022) for a fair comparison. Except for PSNR and SSIM, we also report the results in NIQE (Mittal et al., 2012), Perceutal Index (PI), and Fréchet Inception Distance (FID) (Heusel et al., 2017), which are shown in Table 2. It can be observed that PDN surpasses TGSR by a significant gap on PSNR, SSIM, NIQE, and FID while TGSR outperforms PDN on PI. As pointed out in (Jinjin et al., 2020), PI cannot fairly reflect the subjective performance, since it prefers images with obviously unrealistic artifacts produced by GAN-based methods. Since TGSR is trained in an adversarial setting, it is expected that TGSR poses better PI values. In summary, PDN shows better performance on text-guided image super-resolution than TGSR, which can be explained from four perspectives. First, TGSR's text encoder (LSTM) is trained from scratch along with the main SR network, which cannot take advantage of large-scale multi-modal pre-training. Second, TGSR projects each word into high-dimensional embedding rather than the whole sentence, which neglects the relationship between words. Third, because of the static inference process, TGSR is not capable enough to handle spatial variations, especially when scale factors are large. Last, TGSR uti-

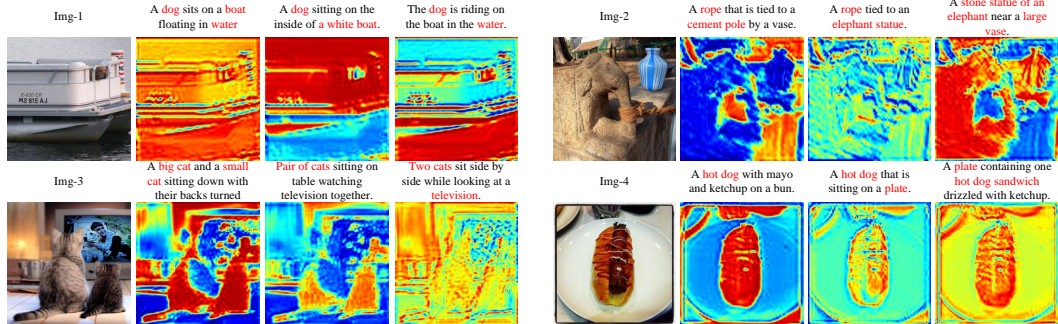

Figure 4: Visualization of multi-modal attention masks given different prompts. The most informative words in the caption are marked red.

lizes an adversarial training strategy which is prone to suffer from geometric distortions (Ma et al., 2020).

Table 2: Quantitative results of TGSR and PDN. The best performance is highlighted in **bold**.

| Method | PSNR↑ | SSIM↑ | NIQE↓ | FID↓ | PI↓ |
|---|---|---|---|---|---|
| Bicubic | 26.59 | 0.810 | 14.514 | 105.232 | 9.676 |
| TGSR | 23.48 | 0.766 | 8.846 | 93.919 | **7.165** |
| PDN | **29.18** | **0.833** | **8.634** | **24.307** | 7.390 |

## 4.4 INCORPORATING DYNAMIC CORRELATION MODULE INTO OTHER MODELS

In this section, we investigate the effect of the proposed Dynamic Correlation Module by incorporating it into EDSR, RDN, and RCAN (Lim et al., 2017; Zhang et al., 2018b;c). The upgraded networks are referred to as EDSR+, RDN+, and RCAN+, respectively. All these upgraded and baseline networks are trained on the COCO dataset with a scale factor of ×4 and the configurations are identical to those described in Section 4.2. The quantitative results are presented in Table 3. It is obvious that the proposed module boosts the SR performances notably. The upgraded networks achieve a PNSR improvement of up to 0.51 dB and at least 0.11 dB. In terms of SSIM, the improvement is 0.02 at the maximum and 0.009 at the minimum, demonstrating the effectiveness of the proposed DCM.

Table 3: Quantitative results of the upgraded networks and the baseline networks. The best performance is highlighted in **bold**.

| Method | Set5 PSNR/SSIM | Set14 PSNR/SSIM | Urban100 PSNR/SSIM |
|---|---|---|---|
| EDSR | 30.74/0.842 | 27.75/0.756 | 24.74/0.726 |
| EDSR+ | **31.13/0.851** | **27.99/0.763** | **25.12/0.740** |
| RCAN | 30.33/0.828 | 27.46/0.744 | 24.34/0.707 |
| RCAN+ | **30.76/0.841** | **27.75/0.755** | **24.77/0.727** |
| RDN | 30.55/0.836 | 27.60/0.751 | 24.55/0.718 |
| RDN+ | **31.06/0.847** | **27.93/0.760** | **25.03/0.737** |

## 4.5 SENSITIVITY TO DIFFERENT PROMPTS

In this section, we present the visualization of multi-modal attention masks given different prompts, as shown in Figure 4. We conduct this experiment on COCO datasets with a scale factor of ×8. Other training details are the same as those described in Section 4.2. It is obvious that multi-modal attention masks tend to assign a large value at the corresponding spatial location which shows high correspondence with the textual description. Taking the image "Img-3" for example, if the caption

is "A *big cat* and a *small cat* sitting down with their backs turned", the cats' areas are greatly highlighted. If the caption changes to "Two *cats* sit side by side while looking at a *television*", the multi-modal attention mask tends to assign large values to the areas of cats and television. More results can be found in the supplementary material.

### 4.6 ABLATION STUDY OF DYNAMIC CORRELATION MODULE

In this section, we present the quantitative analysis of the Spatially Multi-Modal Attention Module and Prompt-Guided Dynamic Convolution Module. Specifically, we remove one of them from the network upgraded with DCM, which is referred to as EDSR+. We denote the EDSR+ without Prompt-Guided Dynamic Convolution Module (replaced with standard convolution) as "EDSR+ w/o dy", and the EDSR+ without Spatially Multi-Modal Attention Module as "EDSR+ w/o att". Then such networks are trained on the COCO dataset with a scale factor of ×4. Table 4 shows the quantitative results.

Table 4: Quantitative results of the ablation studies of DCM. The best performance is highlighted in **bold**.

| Method | Set5 PSNR/SSIM | Set14 PSNR/SSIM | Urban100 PSNR/SSIM |
|---|---|---|---|
| EDSR | 28.94/0.789 | 26.47/0.715 | 23.45/0.670 |
| EDSR+ w/o dy | 29.77/0.814 | 27.04/0.734 | 23.93/0.692 |
| EDSR+ w/o att | 29.81/0.816 | 27.07/0.735 | 23.96/0.692 |
| EDSR+ | **29.95/0.822** | **27.07/0.739** | **24.11/0.701** |

### 4.7 VISUALIZATION OF MULTI-MODAL ATTENTION MASK

Furthermore, to validate the effectiveness of DCM, we provide examples of visualization of attention masks along with their captions and corresponding HR images, as shown in Figure 5. Evidently, multi-modal attention masks generated by DCM not only keep consistency with the most informative words in corresponding captions but also plausibly reflect spatial variations. For instance, when taking the caption "A *tennis player* running and looking up to find the ball" as the prompt, DCM tends to highlight the woman's body area. In summary, we conclude that the multi-modal attention mask enables satisfactory generalization and adaptation of scene descriptions, which then improves the SR performances.

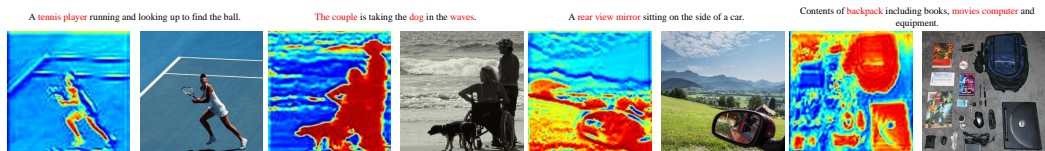

Figure 5: Visualisation of the multi-modal attention masks along with their captions. The most informative words in the caption are marked red.

## 5 CONCLUSION

This paper proposes a Prompt-guided Dynamic Network (PDN) and Dynamic Correlation Module (DCM) for single image super-resolution (SISR), which introduces powerful multi-modal representations into the super-resolve procedure. Specifically, the Spatially Multi-Modal Attention Module is proposed to extract discriminative features according to multi-modal prompts, and the Prompt-Guided Dynamic Convolution Module is the first to introduce multi-modal prompts into convolutional kernel estimation for feature transformation, leading to better cross-modal coherence and spatial variations. Extensive experimental results show that DCM not only boosts the SR performances of existing image SR methods but retains high relevance with textual and visual prompts.

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

## A    APPENDIX

You may include other additional sections here.

