# 1 MORE VISUAL RESULTS ON SINGLE IMAGE SUPER-RESOLUTION

In this section, we provide more visual comparisons with state-of-the-art SR methods, including EDSR, RDN, RCAN, SRGAN, and ESRGAN. As shown in Figure 1, the proposed RDN not only produces high-quality super-resolved images but also preserves the textural details well.

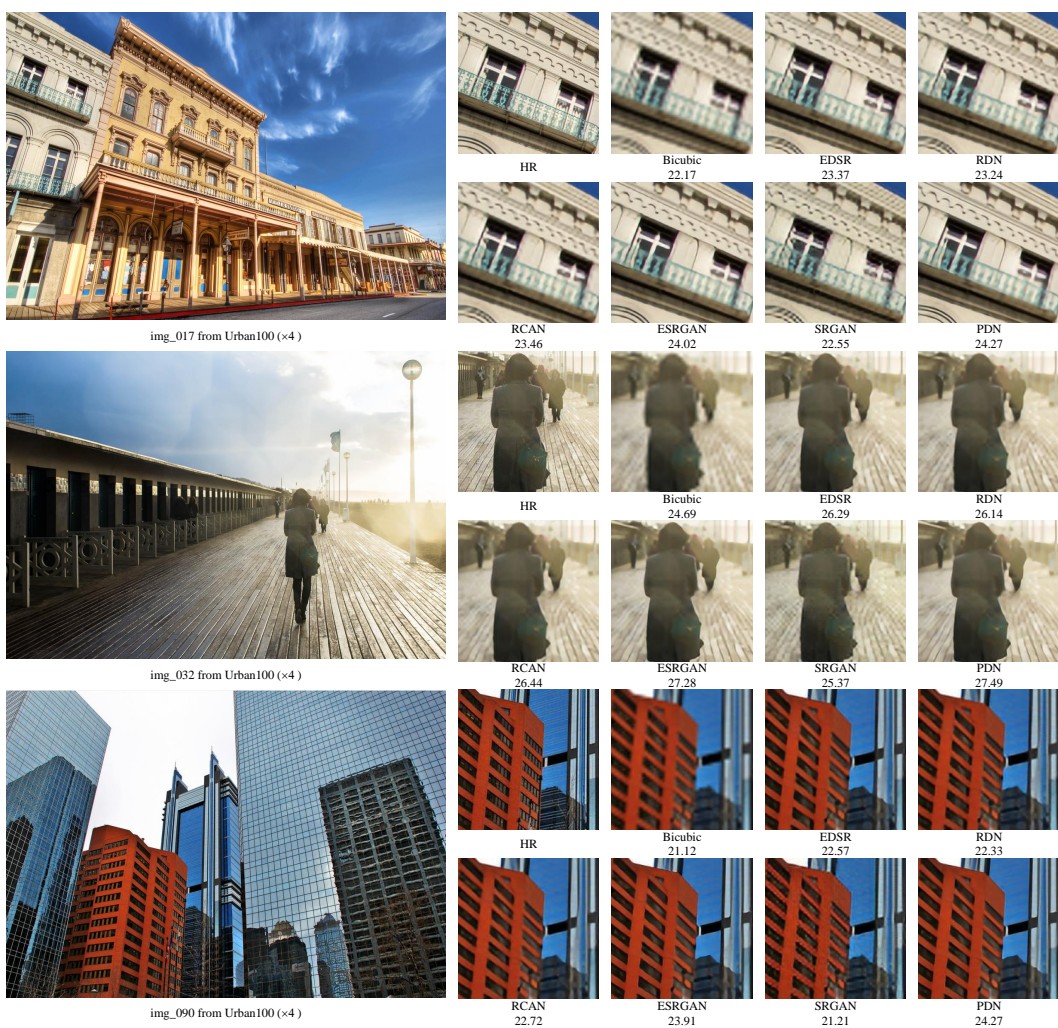

Figure 1: Visual comparison between the proposed RDN and state-of-the-art SR methods. Best viewed on screen.

# 2 VISUALIZATION OF THE MULTI-MODAL ATTENTION MASKS

In this section, we provide more visualization of multi-modal attention masks given different prompts, as shown in Figure 2. It can be observed that the attention masks are highlighted adaptively to the given prompt. Then the SR networks will better concentrate on these areas, leading to the elevation of SR performances.

# 3 STUDY OF PROMPT'S MODALITY

To validate the robustness of DCM to different modalities during the training procedure, we provide visual results of the PDN and state-of-the-art SR methods trained on the FFHQ dataset, as shown in Figure 3. From Figure 3, it can be observed that PDN's results are more authentic and natural

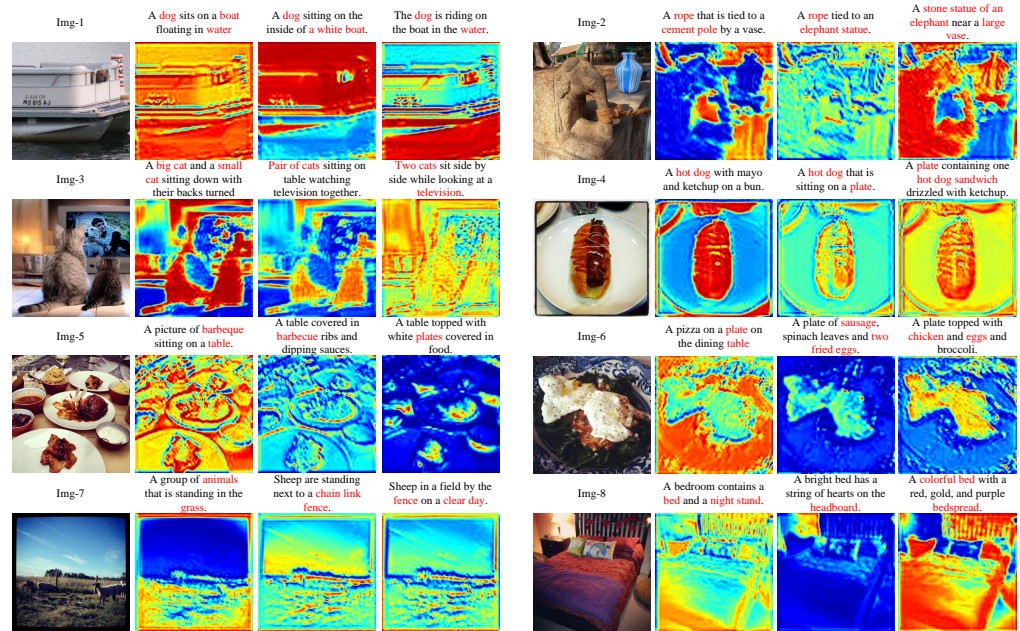

Figure 2: Visualization of the multi-modal attention masks given different text descriptions as prompts. The most relevant words in the descriptions are marked red.

compared with other results. Taking the image "008582" for example, EDSR fails to restore eyeballs clearly; RDN, RCAN, and ESRGAN generate results with perceptually unnatural and blurry forehead; SRGAN even fails to restore the correct skin color. In contrast, with the help of DCM, PDN is capable of generating vivid facial features and poses the highest PSNR value. One may doubt if using a horizontally flipped image as the prompt will leak the high-resolution information to the SR model, but we would like to highlight that the prompt is only used to generate the prompt embedding with CLIP but not fed into the CNN. The prompt embedding has a very high degree of abstraction (i.e., a $d$-dimensional vector), which contains no low-level information. Otherwise, such leakage should have enabled PDN to easily recover the tiny details like freckles (which is impossible under the $\times 8$ setting for a regular SR model), but this is not observed. More importantly, such experiments with image prompts are aimed at verifying PDN's ability to understand prompts other than texts but not chasing the state of the art, which are complementary to the experiments with textual prompts (i.e., those trained on COCO). More results can be found in the supplementary material.

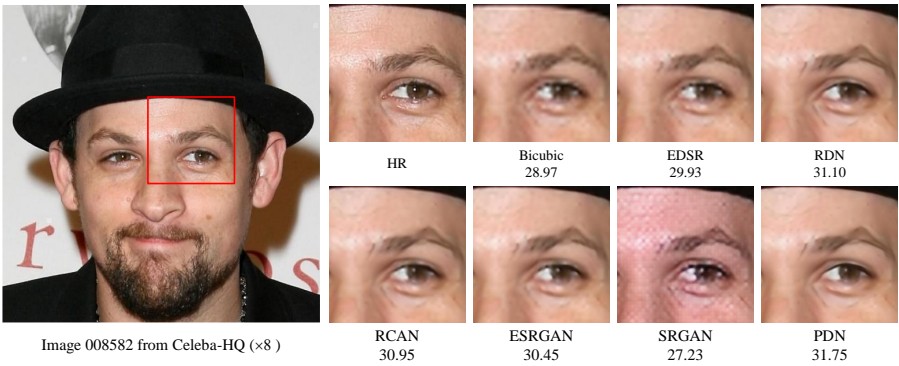

Figure 3: Visual comparison with state-of-the-art SR methods on Celeba-HQ with a scale factor of $\times 8$. Best viewed on screen.

## 4 FAILURE CASES

We provide some examples of confusing and caption-irrelevant multi-modal attention masks in Figure 4. It can be observed that these examples contain unclear main objects. For example, in the bottom-right image, the main scenes are green hills rather than "A bunch of animals" in the corresponding caption. Besides, these images tend to generate image representations with disordered and indivisible features, which will deteriorate the precision of attention masks. In such a scenario, the proposed Multi-Modal Attention Module mistakes the areas with the highest intensity as the most similar areas instead of the most caption-relevant ones.

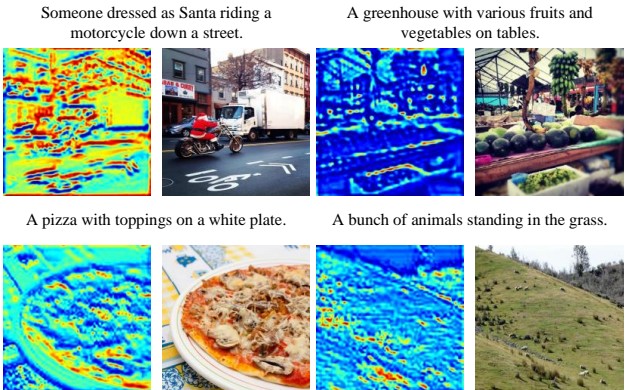

Figure 4: Examples of confusing multi-modal attention masks given their text descriptions.