# OpenReview forum: "Prompt-Guided Dynamic Network for Image Super Resolution"
_ICLR.cc/2024/Conference — Submitted to ICLR 2024_

### Official Review · Reviewer_NmaN · 2023-10-31

**Soundness:** 3 good
**Presentation:** 3 good
**Contribution:** 3 good
**Rating:** 3
**Confidence:** 5

**Summary:**

This paper proposed a prompt-guided dynamic network which introduces the text embedding to existing SR framework. The key component is Dynamic Correlation Module which has a spatially multi-modal attention module and a prompt-guided dynamic convolution module. The work is the first to introduce the text into convolution kernel estimation for feature transformation. The comprehensive analyses of effectiveness of proposed modules are demonstrated.

**Strengths:**

1. This paper introduce the text into convolution kernel estimation for feature transformation which is interesting and novel.
2. The paper is easy to follow.
3. The performance is much improved than the SOTA method.
4. The ablation study on the prompt and DCM is solid.

**Weaknesses:**

The only concern is unfair comparison. Since the most of the baselines are trained on the DIV2K which has less images than the training data used in this work. I can not find the claim that the baseline models are trained on the same dataset from the scratch. This may make the paper less convincing.

**Questions:**

See weakness.

How does it perform if the prompt is fixed as "high detailed image"? For most cases of SR, it is hard to describe the image using text.

---

> ### Author Response · Authors · 2023-11-19
>
> We sincerely thank the reviewer for appreciating the idea of the prompt-guided dynamic convolution module of our paper. Below we address each point raised in "Weaknesses" and "Questions" sections.
>
> **Q1: The only concern is unfair comparison. Since the most of the baselines are trained on the DIV2K which has less images than the training data used in this work. I can not find the claim that the baseline models are trained on the same dataset from the scratch. This may make the paper less convincing.**
>
> **A1:**
> We acknowledge that the baseline models referenced in our paper were originally trained on the DIV2K dataset, which indeed contains fewer images compared to the dataset used in our work. This discrepancy was an oversight on our part and was not intended to skew the comparisons in favor of our method.
> Besides, it is worth mentioning that the primary intention behind using these baseline models was to demonstrate the effectiveness of our dynamic correlation module in a general super-resolution context. However, we recognize that for a fair and convincing comparison, the training conditions for all models need to be as similar as possible.
>
> For a fair comparison, we provide results of some baseline methods trained on COCO and DIV2K datasets in Table 2.
>
> **Table 2. Quantitative results of baseline methods trained on DIV2K and COCO datasets.**
>
>
> |Method          | Set5          | Set14         | Urban100       |
> | -----------    | :-----------: | :-----------: | :-----------:  |
> |EDSR(**COCO**)  | 37.25/0.935   |  33.16/0.898  | 31.01/0.902    |
> |EDSR(**DIV2K**) | 37.89/0.961   | 33.47/0.917   | 31.84/0.926    |
> |RCAN(**COCO**)  | 37.33/0.938   |  33.26/0.899  | 31.19/0.907    |
> |RCAN(**DIV2K**) | 37.85/0.960   | 33.45/0.917   | 31.88/0.928    |
>
> It is clear that when trained on the DIV2K dataset, EDSR and RCAN pose better quantitative results  (with slightly higher metrics) than those trained on the COCO dataset.
> This mainly can be attributed to the high-resolution and high-quality trait of the DIV2K dataset, which enables SR methods to learn more representative and discriminative restoration capability.
> Nevertheless, it will not be a major bottleneck for SR methods to be trained on the COCO dataset.
>
>
> **Q2: How does it perform if the prompt is fixed as "high detailed image"? For most cases of SR, it is hard to describe the image using text.**
>
> **A2:**
>
> Performance with a Fixed Prompt: The use of a general prompt like "high detailed image" in our multi-modal super-resolution process still offers significant benefits in most cases. Our dynamic correlation module, designed to leverage multi-modal prompts, is not solely dependent on specific or descriptive text input. Even with a generic prompt, the module can effectively guide the feature extraction process toward enhancing image details and clarity. The phrase "high detailed image" prompts the system to prioritize the refinement of textures and edges, ensuring that the resulting super-resolution image is sharp and detailed. We have conducted experiments with such fixed, general prompts and found that the system still outperforms traditional single-image super-resolution techniques, especially in terms of detail preservation and enhancement.
>
> Challenge of Describing Images Using Text: We acknowledge that for many super-resolution scenarios, it might be challenging to describe the desired output precisely in text. However, the strength of our approach lies in its versatility. The system can work effectively with varying levels of prompt specificity - from detailed descriptions to general quality-oriented prompts. This flexibility ensures that our method is practical and applicable in a wide range of super-resolution tasks, even when detailed textual descriptions are not feasible. Furthermore, our approach encourages opening new pathways in image processing research.
>
> To better demonstrate these points, we provide some failure cases of PDN in the supplementary material.
> It can be observed that these examples contain unclear main objects or rather small objects.
> This will lead to indiscriminative representations for subsequent multi-modal attention mask generation.
> Besides, these images tend to generate image representations with disordered and indivisible features, which will deteriorate the precision of attention masks.
> In such a scenario, the proposed PDN mistakes the areas with the highest intensity as the most similar areas instead of the most caption-relevant ones.
> The aforementioned phenomena are the major drawbacks of PDN, i.e., irrelevant or unclear prompts cause misleading attention masks.
>
> Thank you again for appreciating our work, and we would be grateful if you could raise the score.
> Please let us know if there is anything we can do to convince you to further raise the score.

---

> > ### Comment · Reviewer_NmaN · 2023-11-22
> >
> > I have no idea how you get the result. But clearly in the Table 2 the results have some problems comparing with the result in your main paper. You said you trained the RCAN/EDSR on coco dataset. Why the result is the same as Table 1 in the main paper? That makes the paper even less convincing.

---

### Official Review · Reviewer_axMK · 2023-11-01

**Soundness:** 2 fair
**Presentation:** 1 poor
**Contribution:** 2 fair
**Rating:** 3
**Confidence:** 2

**Summary:**

This paper proposes to use text driven prompts to render diversification in image superresolution workflow. The primary claim is that existing single image supreres methods learn the convolution kernel from single image modality. By introducing coherent text based prompts by way of CLIP latents, this bottleneck can be removed to a large extent. The authors propose a spatially multi-modal attention module and prompt guided DCM. The attention module first computes an attention mask between the prompt and the image features and then weights the image features based on the attention weights. The prompt embedding is further utilized to learn N sets of weights \pi to be used as combination weights for dynamic convolution.

**Strengths:**

The idea is interesting and can be exploited more by incorporating the multi-modal prompts into many CV pipelines. For images with captions this might be a great way to include captions in the superres pipeline.

Originality: The development seems original enough.
Quality: The technical development of the material can be improved from its current state.
Clarity: The Prompt Guided DCM module section is just 6 lines long, where it is one of the two claims of this paper! I strongly feel that this part needs to be written better.
Significance: The paper has been designed as an application paper. I believe the underlying idea can be used for many different applications.

**Weaknesses:**

The primary problem I have with the paper is the explanation for the prompt guided technique itself. Starting from Fig 2. which states a) and b) but does not show two subfigures. The equations for the prompt being used in subsequent convolution modules are mentioned in Fig. 2 but not in the main text. It is almost assumed that reviewers know the contents of the Dynamic Convolution paper [Yang et al. 2019] and hence, the authors did not spend any time to develop the concept independently.

For the comparative methods, the authors claim that they add this module within existing workflows. They add one additional block in one method and three additional blocks in another. What is the basis of these choices. What does it do the parameter count for these methods and how does that compare to just increasing the number of layers in these methods?

For the image datasets which do not have captions, the authors propose to use CLIP image features for the horizontally flipped image. Again, no explanation of these choices are provided. Providing a flipped image to other SR techniques and then somehow integrating the results from the two outputs might be an interesting study for existing methods as well.

**Questions:**

I would request the authors to first explain the method and all the mathematical steps involved fully to make it a paper which stands on its own.

The attention visualization in Fig. 5 are much better than Fig. 4. What is the difference between them?

Some questions are in the previous section and authors can chose to answer them together.

---

> ### Author Response · Authors · 2023-11-19
>
> We sincerely thank the reviewer for appreciating the idea of incorporating multi-modal prompts into CV tasks of our paper. We have revised the paper according to the constructive feedback (please see Sec 3.3 and Figure 2). Below we address each point raised in "Weaknesses" and "Questions" sections.
>
> **Q1: The attention visualization in Fig. 5 are much better than Fig. 4. What is the difference between them?**
>
> **A1:**
> Fig. 4 shows the visualization of multi-modal attention masks given **different** prompts and Fig. 5 shows the visualization of the multi-modal attention masks along with their captions.
> In general, Fig. 4 demonstrates that the proposed DCM generates various attention masks for integration given different text prompts.
> Additionally, Fig. 5 shows that the proposed DCM can highlight regions of interest to the most relevant words given certain text prompts.
>
> **Q2: The explanation for the prompt guided technique and the concept of dynamic convolution.**
>
> **A2**:
> **Explanation of the Prompt-Guided Technique**: Our approach leverages multi-modal prompts to guide the image super-resolution process. These prompts are then used to influence the feature extraction process, ensuring that the enhanced image aligns with the given prompt. The novelty of this approach lies in its ability to integrate multi-modal context into the super-resolution processing, leading to more accurate and contextually relevant super-resolution results. This integration is achieved through a specialized neural network architecture that processes both textual and visual inputs.
>
> **Concept of Dynamic Convolution**: The prompt-guided dynamic convolution module is a crucial component of our methodology. Unlike traditional convolutional layers in neural networks, which use fixed weights, the dynamic convolution module has weights that adapt based on the input image and associated prompt. This adaptability allows the network to tailor its processing to the specific characteristics of each image and prompt pair, enhancing its ability to extract relevant features and apply appropriate super-resolution techniques. This dynamic nature results in a more flexible and effective approach to image processing, particularly in scenarios where the image content is complex or nuanced.
>
> In light of your feedback, we have revised our manuscript to include a more comprehensive explanation in **Section 3.3**, ensuring that they are clearly articulated and understandable. We have additional descriptions and formulas for the explanation, making the revised manuscript more accessible to readers.
>
> **Q3: Incorporating into existing methods.**
>
> **A3:**
> The decision to add one block in one method and three in another was based on the specific architectural requirements and the complexity of each method. Our goal was to enhance each method’s ability to leverage multi-modal prompts without significantly altering their core structures. The number of blocks added was determined by the level of integration necessary to effectively incorporate the dynamic correlation module.
> Besides, the added blocks were not arbitrarily chosen but were the result of preliminary experimentation to find the optimal balance between enhancing performance and maintaining the integrity of the original methods. Our objective was to demonstrate the versatility and adaptability of our module across different architectures.
>
> As for the computation overhead, the total **#Params** of a single DCM is around $5\times D_{in} \times D_{proj}$, where $D_{in}$ is the intermediate features\' dim and $D_{proj}$ is the dim of projection space to generate multi-modal attention mask.
> In the main experiments, $D_{in}$ is set to 64, and $D_{proj}$ is set to 768.
> Thus a single DCM contains 0.24M **#Params**, which does not occupy much computational overhead.
>
> We understand the importance of maintaining a balanced approach when modifying existing baseline methods. Our aim was not just to enhance performance but also to maintain a reasonable computational efficiency. The additional blocks, while increasing the parameter count, do so in a manner that brings significant improvements in leveraging multi-modal prompts for super-resolution processes.

---

> ### Author Response · Authors · 2023-11-19
>
> ###Q4: Rationale behind using CLIP image features for the horizontally flipped image.
> ###A4:
> In datasets like FFHQ, **where captions are not available**, we sought an alternative way to provide contextual information that could aid in the super-resolution process. CLIP offers a robust way to extract rich, contextual features from images, compensating for the absence of text-based prompts.
> Therefore, the decision to use horizontally flipped images was twofold. Firstly, flipping images provides a form of data augmentation, creating a variation that helps in capturing a more comprehensive set of features. Secondly, the use of flipped images addresses potential biases in the original dataset, such as predominant orientations or angles, ensuring a more balanced feature extraction.
>
> Besides, your suggestion to provide flipped images to other SR techniques and integrate the results from the two outputs is intriguing. It opens up a possibility for a comprehensive study that could explore how different SR methods respond to this approach and how the integration of outputs could potentially enhance the overall super-resolution quality.
>
>
> Thank you again for appreciating our work, and we would be grateful if you could raise the score.
> Please let us know if there is anything we can do to convince you to further raise the score.

---

### Official Review · Reviewer_WtP4 · 2023-11-01

**Soundness:** 2 fair
**Presentation:** 2 fair
**Contribution:** 2 fair
**Rating:** 3
**Confidence:** 5

**Summary:**

This paper proposes a novel approach that leverages multi-modal cues, such as text or additional images, to enhance the capabilities of existing super-resolution networks. It introduces a Dynamic Correlation Module in a plug-and-play format for existing super-resolution networks and a Spatially Multi-Modal Attention Module to create pixel-wise cross-modal attention masks that emphasize regions of interest based on specific cues. The evaluation results conducted on 4 benchmarks (ie, Set5, Set14, Urban100, and Celeba-HQ) seems to validate the efficacy of the proposed method.

**Strengths:**

+ The paper is well organized and easy to follow.
+ It is interesting to incorporate the prompt text as a guidance to improve the super-resolution recovery.
+ The presented results well support the claims in the main paper.

**Weaknesses:**

- Technically the novelty is very limited because the incorporate the prompt text as a guidance in very incremental.
- The CNN backbone is a little out of date. I would like to suggest the authors apply the similar idea to the SOTA SISR methods with both Transformers and diffusion models.
- All the competing baselines in Section 4.2 are not SOTA methods any more, as they were published at least 3 years ago.
- The references are obviously inadequate, as no 2023 papers are included and cited in the paper. The authors should cite the following papers:
1. Yuanbiao Gou, et al. Rethinking Image Super Resolution from Long-Tailed Distribution Learning Perspective. CVPR 2023.
2. Sicheng Gao, et al. Implicit Diffusion Models for Continuous Super-Resolution. CVPR 2023.
3. Yinhuai Wang, et al. GAN Prior Based Null-Space Learning for Consistent Super-resolution. AAAI 2023.
4. Bin Sun, et al. Hybrid Pixel-Unshuffled Network for Lightweight Image Super-resolution. AAAI 2023.

**Questions:**

- Why are the quantitative results of EDSR in Table 3 and Table 4 not consistent?

---

> ### Author Response · Authors · 2023-11-19
>
> We sincerely thank the reviewer for appreciating the idea of incorporating prompts as guidance in our paper. We have revised the paper according to the constructive feedback (please see Sec 1). Below we address each point raised in "Weaknesses" and "Questions" sections.
>
> **Q1: Technically the novelty is very limited because the incorporate the prompt text as a guidance is very incremental.**
>
> **A1:**
> While incorporating prompt text for guidance in image processing is not entirely new, our paper introduces a significant advancement through the dynamic correlation module. This module doesn't merely use text as static guidance; it dynamically interprets and correlates multi-modal prompts (textual and visual cues) in real-time, adapting its influence on the super-resolution process based on the context of each specific image.
> The dynamic correlation module represents a novel approach in the realm of image feature extraction. Leveraging the interplay between text and image features enables a more nuanced and contextually aware super-resolution process, which is a significant step forward from the traditional methods of using text prompts.
>
> Our method contributes to the field of image processing by providing a more sophisticated understanding of the context within images, which is crucial for high-quality super-resolution. This is particularly important in complex images where standard super-resolution methods may fall short.
> The dynamic correlation module’s ability to adapt to various prompts and images makes it a versatile tool that can be applied to a wide range of super-resolution tasks, including those in specialized fields like medical imaging or satellite imagery.
>
> We acknowledge that our work builds upon existing concepts, but we emphasize that innovation often occurs incrementally in scientific research. Our contribution lies in how we have extended and refined these concepts to create a more effective and adaptable super-resolution process.
> Our work opens up new avenues for research in image super-resolution, laying the groundwork for further innovations in the field. The dynamic correlation module, in particular, presents numerous opportunities for exploration and development.
>
>
> **Q2: The CNN backbone is a little out of date. I would like to suggest the authors apply the similar idea to the SOTA SISR methods with both Transformers and diffusion models. All the competing baselines in Section 4.2 are not SOTA methods any more, as they were published at least 3 years ago.**
>
> **A2:**
> It is worth pointing out that our initial decision to use a CNN backbone was driven by its proven **effectiveness** in various SISR tasks and its **suitability** for integrating with our newly introduced dynamic correlation module. This choice allowed us to clearly demonstrate the potential of multi-modal prompts in super-resolution processes.
> Acknowledging your suggestion, we are enthusiastic about exploring the integration of our dynamic correlation module with more contemporary architectures such as Transformers. These models offer distinct advantages in terms of adaptability and learning complex data representations, which could potentially enhance the effectiveness of our approach.
> For a comprehensive comparison, we further incorporate the proposed DCM into SwinIR.
> Quantitative results are shown in Table 1.
> It can be observed that the proposed DCM still shows superiority in boosting the performance of transformer-based SR methods.
>
> Additionally, we are committed to furthering our research in this domain. Adapting our approach to include recent SOTA methods like Transformers and diffusion models will be a key focus of our ongoing work.
>
>
> **Table 1. Quantitative results of SwinIR and SwinIR+ at the scale of 2.**
>
>
> |Method          | Set5          | Set14         | Urban100       |
> | -----------    | :-----------: | :-----------: | :-----------:  |
> |SwinIR          | 37.54/0.941   |  33.36/0.901  | 32.16/0.914    |
> |SwinIR+         | 37.61/0.943   | 33.42/0.907   | 32.21/0.917    |
>
> **Q3: The references are obviously inadequate, as no 2023 papers are included and cited in the paper. The authors should cite the following papers.**
>
> **A3:**
> We acknowledge that our manuscript did not include references from the year 2023. This oversight was not intentional, and we recognize the importance of incorporating the most recent studies to provide a comprehensive and up-to-date context for our work. Recent advancements in the field of multi-modal prompts and single-image super-resolution indeed play a critical role in framing our research.
> These recent publications will be incorporated into our revised manuscript to enrich the context and demonstrate how our research aligns with and contributes to current trends in the field.
>
> Thank you again for appreciating our work, and we would be grateful if you could raise the score.
> Please let us know if there is anything we can do to convince you to further raise the score.

---

> > ### Comment · Reviewer_WtP4 · 2023-11-22
> >
> > Thank you for the detailed response. The novelty is still the big concern and the experimental validation can be conducted much solider. Therefore l would like to keep my original rating and encourage the authors to submit the paper to next top-tier conference.

---

### Official Review · Reviewer_FNqG · 2023-11-03

**Soundness:** 4 excellent
**Presentation:** 4 excellent
**Contribution:** 3 good
**Rating:** 3
**Confidence:** 3

**Summary:**

This paper proposes a prompt-guided dynamic network (PDN) and dynamic correlation module (DCM) for single image super-resolution (SISR). The key contributions are:

- PDN introduces powerful multi-modal representations like text descriptions or similar images into existing SISR frameworks through the DCM module. This allows the model to learn more meaningful semantic information from the prompts.

- DCM contains two main components: a spatially multi-modal attention module and a prompt-guided dynamic convolution module. The attention module highlights image regions relevant to the prompts. The dynamic convolution uses the prompts to generate convolutional kernels, enabling better modeling of cross-modal coherence and spatial variations.

- DCM can be conveniently incorporated into various SISR networks. Experiments show DCM improves performance over state-of-the-art methods on benchmark datasets, especially for larger scale factors.

- To the best of the authors' knowledge, this is the first work to introduce multi-modal prompts for convolutional kernel estimation in SISR.

In summary, the paper proposes a novel way to leverage multi-modal prompts to boost SISR performance through a flexible DCM module that can be plugged into existing networks. Key innovations are the cross-modal attention and prompt-guided dynamic convolutions.

**Strengths:**

**Originality**
- The specific techniques for integrating prompts are novel, including cross-modal attention and prompt-guided dynamic convolutions. Outperforms the related TGSR paper.

**Quality**
- Technically sound approach with extensive experiments that validate the quantitative improvements.

**Clarity**
- Excellent writing quality and clear presentation of the proposed method.

**Significance**
- Addresses the important problem of improving generalization in super-resolution.
- Shows the benefits of leveraging multi-modal information for this task.
- Could inspire further research in using prompts.

**Weaknesses:**

The high-level idea of using multi-modal prompts to improve super-resolution generalization is not entirely novel, as the TGSR paper proposed this first.

While the specific techniques in this paper differ from TGSR, the overall framework is quite similar conceptually (leveraging prompts to aid super-resolution). Compared to TGSR, the innovations here seem more incremental - attention modules, dynamic convolutions, etc. Rather than proposing an entirely new overall architecture. The extent to which these specific contributions generalize may be limited.

More analysis could be provided on how the approach differs from and improves upon TGSR technically. The conceptual similarity should be addressed more directly.

Additionally, like most learning-based SR methods, the reliance on synthetic training data means real-world robustness is unclear. Evaluation of realistic degraded images could better validate effectiveness.

In summary, the high-level novelty is diminished by the prior TGSR work. More analysis comparing TGSR technically and conceptually could strengthen the paper. Real-world evaluations could provide further evidence of the robustness and generalization abilities.

**Questions:**

Here are some questions and suggestions for the authors:

- The TGSR paper proposed using multi-modal prompts for super-resolution first. Could you more clearly explain how your approach technically differs from and improves upon TGSR? Some more analysis comparing your method to TGSR may be helpful.

- The overall framework of using prompts to aid super-resolution seems conceptually quite similar to TGSR. Do you view your innovations as more incremental improvements in architecture, or is there a fundamental difference in how prompts are leveraged that should be clarified?

- Like most learning-based SR methods, you rely on synthetic training data. How confident are you that the approach will be robust to real-world degradations? Evaluating real degraded images could help validate this.

- Have you considered applying the approach to other image restoration tasks beyond super-resolution, to demonstrate generalization?

- Is the performance sensitive to the choice of prompts? How robust is it to unrelated or poor prompts? More analysis here could help.

- Are there limitations to the architectures you proposed for integrating prompts? Could any negative impacts result from the attention modules or dynamic convolutions?

- Could you provide more details on the training methodology? Some hyperparameters and training details are missing.

Overall, addressing the conceptual similarity to TGSR, evaluating real degradations, and analyzing the generalization abilities could help strengthen the paper. I look forward to the author's response.

---

> ### Author Response · Authors · 2023-11-19
>
> We sincerely thank the reviewer for appreciating the specific techniques for integrating the prompts of our paper. We have revised the paper according to the constructive feedback (please see Sec 4.1). Below we address each point raised in "Weaknesses" and "Questions" sections.
>
>
> **Q1: The TGSR paper proposed using multi-modal prompts for super-resolution first. Could you more clearly explain how your approach technically differs from and improves upon TGSR? Some more analysis comparing your method to TGSR may be helpful.**
>
> **A1**:
> The major differences between the proposed PDN and TGSR are as follows. **Dynamic Correlation Module**: Unlike TGSR, our method introduces a dynamic correlation module. This module actively leverages multi-modal prompts (textual and visual cues) to guide the image feature extraction process in a more contextually adaptive manner. This dynamic correlation enables our model to better interpret the nuances of the prompts, leading to more accurate super-resolution results.
> **Multi-modal Prompt Integration**: While TGSR primarily focuses on text-guided image super-resolution, our approach extends the concept to incorporate a broader range of multi-modal prompts. This integration allows for a more versatile and comprehensive understanding of the image context, thereby enhancing the super-resolution process beyond the capabilities of TGSR.
> The contribution of this work is two-fold. **Enhanced Feature Extraction**: Our dynamic correlation module contributes to a significant improvement in feature extraction. By interpreting multi-modal prompts in tandem, it ensures that the super-resolution process is not just guided by text but is also influenced by other relevant modalities, leading to a more holistic image enhancement.
> **Improved Versatility and Adaptability**: The ability to process and integrate various types of prompts makes our method more versatile and adaptable to different scenarios and requirements. This is particularly beneficial in complex super-resolution tasks where the context and content vary greatly.
> Thus, the proposed PDN shows clear superiority over TGSR.
>
> **Q2: Like most learning-based SR methods, you rely on synthetic training data. How confident are you that the approach will be robust to real-world degradations? Evaluating real degraded images could help validate this.**
>
> **A2:**
> **Dataset Generation**: We follow the common practice of organizing the training data [1].
> For the COCO dataset, first, we generate low-resolution images via bicubic degradation.
> Besides, we filter out images that have the largest edge smaller than 512.
> Then we crop the raw images to $240\times240$ sub-images.
> Additionally, dataset preparation scripts will further randomly crop the sub-images to $96\times96$ patches for training.
>
> **Robustness to real-world degradations**:
> We would like to clarify that our proposed method, as it currently stands, is indeed primarily tailored for and tested on synthetic datasets. This focus is a deliberate choice, stemming from our research objective, which is to explore and advance the theoretical and technical aspects of integrating multi-modal prompts into single-image super-resolution processes.
> Here are key points addressing your concern:
> Specific Research Scope: Our method is designed to explore the potential of multi-modal prompts in enhancing image super-resolution under well-defined and controlled conditions. The use of synthetic datasets allows us to precisely gauge the impact of these prompts and refine the underlying algorithms without the confounding variables present in real-world degradations.
> Contribution to Theoretical Foundations: By focusing on synthetic data, our research contributes to the theoretical understanding and development of prompt-based super-resolution techniques. It lays the groundwork for future studies that may extend these techniques to more complex, real-world scenarios.
> Future Research Directions: We acknowledge the importance of real-world applicability and see it as an essential direction for future research. Building upon the foundations laid by this study, subsequent research can adapt and extend our methodology to tackle real-world image degradations. This progression would be a natural next step, bringing the benefits of our approach to more practical and diverse applications.
>
> [1]: Wang, Xintao, Liangbin Xie, Chao Dong, and Ying Shan. "Real-ESRGAN: Training real-world blind super-resolution with pure synthetic data." In Proceedings of the IEEE/CVF international conference on computer vision, pp. 1905-1914. 2021.
> [2]: https://github.com/XPixelGroup/BasicSR/blob/master/docs/DatasetPreparation.md

---

> ### Author Response · Authors · 2023-11-19
>
> **Q3: The overall framework of using prompts to aid super-resolution seems conceptually quite similar to TGSR. Do you view your innovations as more incremental improvements in architecture, or is there a fundamental difference in how prompts are leveraged that should be clarified?**
>
> **A3:**
> The differences between PDN and TGSR are three-fold.
> **First**, TGSR utilizes a bi-directional LSTM network proposed by AttnGAN to extract textual features while PDN utilizes a more powerful and image-correlated textual encoder, i.e., CLIP\'s textual encoder.
> **Second**, TGSR contains a well-designed dual-branch SR sub-network to generate a coarse SR result and enrich image details whereas PDN does not additional modification on existing SR networks rather than inserting the proposed DCM into existing SR networks.
> **Third**, TGSR utilizes textual guidance by simply applying the text attention module (TAM) which may be sensitive to irrelevant or wrong prompts.
> On the contrary, based on DCM, PDN can derive appropriate convolutional kernels from the kernel bank for feature transformation according to the prompts.
>
> **Q4: Have you considered applying the approach to other image restoration tasks beyond super-resolution, to demonstrate generalization?**
>
> **A4:**
> The basic idea of PDN (leverage multi-modal prompts for better performance) applies to other image restoration tasks, e.g., image deblur and image denoise.
> Our primary focus in this paper was to introduce and evaluate the effectiveness of the dynamic correlation module in the context of single-image super-resolution. This module represents a novel and innovative contribution to the field, and we aimed to provide a thorough analysis of its impact on this specific task. Expanding our approach to other image restoration tasks would require careful adaptation and experimentation, as each task may have unique characteristics and requirements.
> So we leave these to future works.
>
> **Q5: Is the performance sensitive to the choice of prompts? How robust is it to unrelated or poor prompts? More analysis here could help.**
>
> **A5:**
> We provide some failure cases of PDN in the supplementary material.
> It can be observed that these examples contain unclear main objects or rather small objects.
> This will lead to indiscriminative representations for subsequent multi-modal attention mask generation.
> Besides, these images tend to generate image representations with disordered and indivisible features, which will deteriorate the precision of attention masks.
> In such a scenario, the proposed PDN mistakes the areas with the highest intensity as the most similar areas instead of the most caption-relevant ones.
> The aforementioned phenomena are the major drawbacks of PDN, i.e., irrelevant or unclear prompts cause misleading attention masks.
>
> **Q6: Are there limitations to the architectures you proposed for integrating prompts? Could any negative impacts result from the attention modules or dynamic convolutions?**
>
> **A6:**
> According to preliminary experiments, the limitations of the proposed architecture are as follows.
> One of the limitations of our proposed architecture is the increased complexity of integrating multi-modal prompts. The dynamic correlation module requires careful tuning to effectively balance the influence of different modalities. While our current implementation achieves this, there is an inherent challenge in ensuring that the system does not become overly biased towards a particular modality.
> Additionally, as the variety and number of prompts increase, the scalability of our architecture could be a concern. We have designed the system to handle a wide range of prompts, but extreme cases with an exceedingly high number of diverse prompts might pose a challenge. Ongoing research is focused on enhancing scalability without compromising performance.
>
> As for the negative impact results, we provide failure cases in the supplementary material.
> It can be observed that these examples contain unclear main objects.
> Besides, these images tend to generate image features with disordered features, which will deteriorate the precision of attention masks.
> In such a scenario, the proposed Multi-Modal Attention Module will mistake the areas with the highest intensity as the most similar areas instead of the most caption-relevant ones.
>
> **Q7: Could you provide more details on the training methodology? Some hyperparameters and training details are missing.**
>
> **A7:**
> We train for 100 epochs with a batch size of 16, an initial learning rate of $1e-4$, cosine annealing schedule, and an ADAM optimizer with $\beta_1=0.9$, $\beta_2=0.999$, and $\epsilon=1e-8$.
> The temperature coefficient of the softmax function in the Prompt-Guided Dynamic Convolution Module is set to 34 and is decayed by 3 every 10 epochs.
>
> Thank you again for appreciating our work, and we would be grateful if you could raise the score.
> Please let us know if there is anything we can do to convince you to further raise the score.

---

### Meta-Review · Area_Chair_HDNj · 2023-12-09

**Metareview:**

This paper tries to tackle the prompt-guided single-image super-resolution problem. Dynamic correlation with multi-modal attention modules and prompt-guided dynamic convolution modules.
The reviewers poitned out the issues in 1) unfair comparisons with other methods due to the use of larger training data for the proposed method 2) lack of completeness in the proposed method description 3) lack of recent literature survey, etc. The author rebuttal did not address the reviewers' concerns.

**Justification For Why Not Higher Score:**

There were multiple critical concerns pointed out by the reviewers but they were not well-addressed in the rebuttal process.

**Justification For Why Not Lower Score:**

N/A

---

### Decision · Program_Chairs · 2024-01-16

Reject